# Observation of solid-state bidirectional thermal conductivity switching in antiferroelectric lead zirconate (PbZrO₃)

Kiumars Aryana [1], John A. Tomko [1], Ran Gao[2], Eric R. Hoglund [3], Takanori Mimura[3], Sara Makarem[3], Alejandro Salanova[3], Md Shafkat Bin Hoque [1], Thomas W. Pfeifer[1], David H. Olson[1], Jeffrey L. Braun[1], Joyeeta Nag[4], John C. Read[4], James M. Howe[3], Elizabeth J. Opila [1,3], Lane W. Martin [2,5], Jon F. Ihlefeld[3,6 ✉] & Patrick E. Hopkins [1,3,7 ✉]

Materials with tunable thermal properties enable on-demand control of temperature and heat flow, which is an integral component in the development of solid-state refrigeration, energy scavenging, and thermal circuits. Although gap-based and liquid-based thermal switches that work on the basis of mechanical movements have been an effective approach to control the flow of heat in the devices, their complex mechanisms impose considerable costs in latency, expense, and power consumption. As a consequence, materials that have multiple solid-state phases with distinct thermal properties are appealing for thermal management due to their simplicity, fast switching, and compactness. Thus, an ideal thermal switch should operate near or above room temperature, have a simple trigger mechanism, and offer a quick and large on/off switching ratio. In this study, we experimentally demonstrate that manipulating phonon scattering rates can switch the thermal conductivity of antiferroelectric PbZrO₃ bidirectionally by −10% and +25% upon applying electrical and thermal excitation, respectively. Our approach takes advantage of two separate phase transformations in PbZrO₃ that alter the phonon scattering rate in different manners. In this study, we demonstrate that PbZrO₃ can serve as a fast (<1 second), repeatable, simple trigger, and reliable thermal switch with a net switching ratio of nearly 38% from ~1.20 to ~1.65 W m⁻¹ K⁻¹.

[1] Department of Mechanical and Aerospace Engineering, University of Virginia, Charlottesville, VA 22904, USA. [2] Department of Materials Science and Engineering, University of California, Berkeley, Berkeley, CA 94720, USA. [3] Department of Materials Science and Engineering, University of Virginia, Charlottesville, VA 22904, USA. [4] Western Digital Corporation, San Jose, CA 95119, USA. [5] Materials Sciences Division, Lawrence Berkeley National Laboratory, Berkeley, CA 94720, USA. [6] Department of Electrical and Computer Engineering, University of Virginia, Charlottesville, VA 22904, USA. [7] Department of Physics, University of Virginia, Charlottesville, VA 22904, USA. ✉email: jfi4n@virginia.edu; phopkins@virginia.edu

Dynamic control over heat flow in solid-state materials has a wide range of applications from the nanoscale, where enhanced temperature stabilization in electronic devices[1,2] and boosted the efficiency of thermoelectric generators[3,4] has been demonstrated, to the macroscale, where thermal control systems are mandatory for many space-exploration technologies[5–8]. The ability to change the thermal conductivity ($k$) of a material "on-demand" has gained significant traction in recent years, with research efforts focused on materials and mechanisms that enable large on/off switching ratios ($k_{high}/k_{low}$), fast modulation between the two states (<seconds), and trigger mechanisms that can be easily accessed in solid-state architectures (e.g., no moving parts). With developments in materials science and thermometry techniques, several material systems have been discovered with thermal-switching behavior under different stimuli such as electrical[9], thermal[10], electrochemical[11,12], optical[13], magnetic[14], strain[15], and even hydration[16]. Although some of these materials provide large switching ratios (up to an order of magnitude), their complicated trigger mechanisms and the associated switching timescale limit their applications. For instance, most recently, Lu et al.[11] demonstrated the thermal conductivity of $SrCoO_{2.5}$ can be bidirectionally tuned by nearly $10 \pm 4$-fold via electrochemically oxygenating and hydrogenating the film. Although their reported switching ratio is significant, it occurs over a time span of several minutes and degrades over multiple switching cycles. In addition, the use of a liquid/gel electrolyte for enforcing the electrochemical reaction adds further complications to integrating the thermal switch into many device architectures.

A promising class of material candidates for dynamic thermal conductivity switching that has recently received considerable attention due to their fast, repeatable, and well-integrated trigger mechanism are ferroelectric (FE) perovskites, such as lead zirconate titanate $PbZr_{1-x}Ti_xO_3$ (PZT) and lead titanate $PbTiO_3$ (PTO)[9,17,18]. In these FE materials, the application of a sufficiently large electric field alters the ferroelectric domain structure and the corresponding domain-wall population densities. The thermal conductivity of these FE solids is dominated by phonons, and any variations in their domain-wall density could potentially impact their scattering rate and change their thermal conductivity. In this regard, it has been shown that in ferroelectric $BiFeO_3$ the thermal boundary conductance between the domains is lower than that of grain boundaries and results in strong scattering of vibrational modes[19]. Later, Ihlefeld et al.[9] showed the thermal conductivity of $PbZr_{0.3}Ti_{0.7}O_3/PbZr_{0.7}Ti_{0.3}O_3$ bilayers, deposited on silicon substrates moderately decreases by 11% upon application of an electric field; their observation was attributed to an increase in domain-wall density and a corresponding increase in the phonon-boundary scattering rate. In a different approach, Foley et al.[20] showed the thermal conductivity of suspended PZT membranes could be increased by 13% with electric field biasing. In this geometry, the ferroelectric film is not mechanically clamped to the substrate, which allows the domain size to increase and *reduce* the phonon-boundary scattering rate. In another work, Langenberg et al.[15] showed the thermal conductivity of epitaxially grown PTO with high domain-wall density is 61% lower than that of the single-domain film. More recently, through first-principles simulations, Liu et al.[21] showed that the thermal conductivity of PTO can be bidirectionally tuned by applying electric fields of opposite polarities. Although, they attributed the bidirectional thermal switching to a combined change in the unit-cell structure and domain-wall response, previous experimental works on ferroelectric materials such as PZT as well as the current study, found that the thermal conductivity changes uni-directionally upon application of electric field of opposite polarities[9,20].

Despite the large focus on investigating the thermal conductivity of PZT under varying conditions, its antiferroelectric (AFE) end-member, lead zirconate $PbZrO_3$ (PZO), has received much less attention. In this material, a sufficient electric field will transition the antiferroelectric phase (orthorhombic space group *Pbam*) to a ferroelectric phase (rhombohedral space group *R3m*), where there is a volume expansion, reduction in the unit-cell size from 8 formula units to 6, and the possibility of altering the populations of ferroelastic domains[22]. Furthermore, these ferroelastic domains within antiferroelectric PZO may directly impact phonon scattering rates. In addition to the AFE-to-FE phase transition, PZO undergoes another phase transition upon heating, transitioning from antiferroelectric to paraelectric (PE), where the lattice structure goes from 8 formula units to 1 (cubic space group $Pm\bar{3}m$), and thus can be expected to reduce the phonon scattering rate and lead to higher thermal conductivities.

The application of PZO in functional devices relies heavily on the ability to maintain and regulate the material's temperature; for example, both the pyroelectric[23] and electrocaloric[24,25] efficiency of PZO are highly temperature-sensitive quantities. In this work, we aim to fill this void in the literature by investigating both the temperature-dependent thermal conductivity as well as the electric-field dependence of PZO thin films in both polycrystalline and epitaxial embodiments. In particular, we demonstrate how the thermal conductivity of an antiferroelectric solid, PZO, can be bidirectionally switched between low- and high-thermal conductivities using electrical and thermal excitation, respectively. We show that the thermal conductivity of PZO decreases by 10% via domain restructuring during electrical biasing and can be increased by up to 25% upon heating; the combination of these mechanisms allows for thermal conductivity switching of ~38%, which is significantly larger than previously reported switching in PZT and PTO[9,17,20].

## Results

**Sample preparation.** Epitaxial PZO thin films were grown to a thickness of 60 nm on a 15 nm $SrRuO_3$-buffered $DyScO_3$ (110) substrate using established pulsed laser deposition (PLD) procedures detailed in our previous work[26]. 25 $\mu$m diameter with a thickness of 80 nm $SrRuO_3$ contacts were patterned on the PZO film to serve as electrical contacts and transducers for thermal measurements. In addition, polycrystalline PZO films were prepared to a thickness of 300 nm via chemical solution deposition (CSD) on a 100 nm Pt/40 nm ZnO/300 nm $SiO_2$/(001) Si substrate. Au and Al transducer metals were deposited on the polycrystalline PZO with thickness of 80 nm via e-beam evaporation. Figure 1a shows a reciprocal space map taken on the epitaxial film. The film is $(120)_O$ oriented where *O* denotes an orthorhombic phase (i.e., the film $[120]_O$ is parallel to the substrate normal $[110]_O$) and exhibits (expected) 90° structural domains as indicated by the presence of $440_O$ and $280_O$ diffraction peaks. A polarization-electric field, P(E), hysteresis loop is shown for this film in Fig. 1b and reveals antiferroelectric response with an AFE-to-FE transition at ~380 kV/cm. Figure 1c shows the 2$\theta$-$\omega$ XRD pattern for the polycrystalline film. The film is phase-pure without pyrochlore or PbO secondary phases. The higher intensity of the $202/042_O$ reflection suggests that the film has a preferred crystallographic texture. A channeling-contrast backscatter scanning electron micrograph of the polycrystalline film is shown in Fig. 1d. The average grain size is 3.3 $\mu$m and some ferroelastic stripe domains were observed, as indicated by the arrows. The ferroelastic domain-wall spacing varies, but appears to be on the order of 200 nm and are much larger than those in the bilayer PZT films in a prior study on thermal conductivity switching[9]. Figure 1e shows the phase diagram for lead

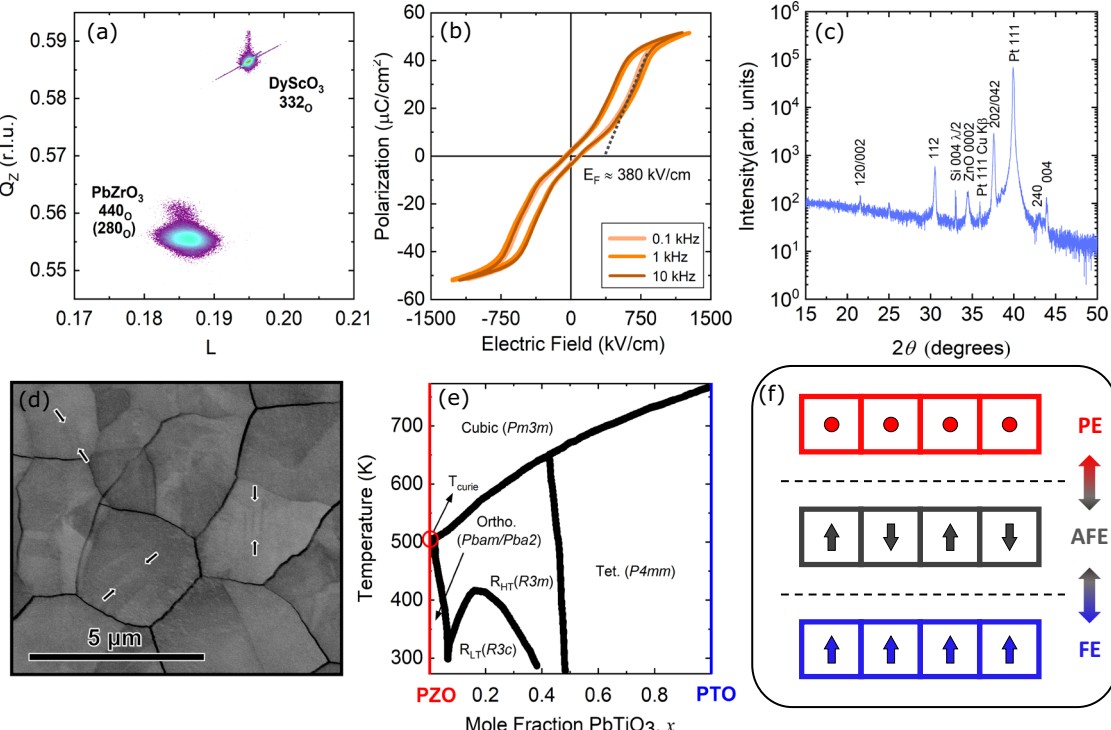

**Fig. 1 Structural phase transformations in PZO upon electrical and thermal stimuli. a** Reciprocal space map of the PbZrO$_3$ 440/280$_O$ and DyScO$_3$ 332$_O$ reflections demonstrating epitaxial growth and the presence of ferroelastic domains in the epitaxial film. **b** Polarization-electric field hysteresis response for the epitaxial PbZrO$_3$ film showing antiferroelectric switching. **c** 2θ-ω XRD pattern for the polycrystalline PbZrO$_3$ film. **d** Channeling-contrast backscatter electron micrograph of the polycrystalline PbZrO$_3$ film. The arrows indicate the locations of clearly resolved ferroelastic domains. **e** Phase diagram for lead zirconate titanate (PbZr$_{1-x}$Ti$_x$O$_3$, PZT) recreated from ref. [27]. **f** Schematic of dipole orientation across antiferroelectric to ferroelectric (AFE-to-FE) and antiferroelectric to paraelectric (AFE-to-PE) phase transitions.

zirconate titanate (PbZr$_{1-x}$Ti$_x$O$_3$, PZT) recreated from ref. [27]. According to this diagram, the PZO would undergo an antiferroelectric to paraelectric phase transition at an elevated temperature near ~500 K. The schematic in Fig. 1f shows the orientation of dipoles across two different phase transitions in PZO upon electrical (AFE-to-FE) and thermal (AFE-to-PE) excitation.

**Temperature rise due to optical heating.** For rapid heating of PZO, we use a continuous wave (CW) laser beam operating at wavelength of 532 nm to deliver large amounts of thermal energy locally. For estimating the temperature profile due to laser heating, precise knowledge of parameters such as heater spot size, deposited power, surface absorption, and the thermal properties of the underlying layers is necessary. For measuring the heater beam size, we use a chalcogenide-based phase-change material, Ge$_2$Sb$_2$Te$_4$ (GST), that undergoes a structural phase transition under thermal excitation. The phase transition from an amorphous to a crystalline phase is initiated at nearly 150 °C where its thermal conductivity increases by almost a factor of three and, upon further heating, can change by almost an order of magnitude[28,29]. By initiating this temperature-induced phase transition in GST with our laser heating source, a permanent spatially dependent thermal conductivity pattern is formed. We then characterize this pattern, and thus determine the laser-heating profile, using a recently-developed thermal conductivity mapping technique[30]. For this, we take a sample with a 40-nm-thick amorphous GST film deposited on a silicon substrate and shine the heater laser with 740 mW power on its surface, which is coated with 60 nm of ruthenium (60 nm Ru/5 nm W/40 nm GST/5 nm W/Si). Subsequently, we spatially map the thermal conductivity of the heated region and based

on the changes in thermal conductivity[31], we determine the spot size of the heater laser. The result of our thermal conductivity map (Fig. 2a) shows a change from ~0.15 W m$^{-1}$ K$^{-1}$ at the unheated area to ~1 W m$^{-1}$ K$^{-1}$ at the center of the beam. By fitting a Gaussian distribution to the thermal conductivity data in Fig. 2b, we measure the heater diameter to be 7.6 μm at 1/$e^2$. To further assess this critical aspect of laser-heating calibration, we also perform knife-edge measurement to determine the beam's spot sizes; we find the two methods are in good agreement.

In order to estimate the temperature distribution profile as a result of laser heating, we turn to finite-element simulations using COMSOL Multiphysics. For this, we use a 2D axisymmetric configuration in cylindrical coordinates (Fig. 2c) with an adiabatic boundary condition at the top surface and an open boundary condition at the sides and bottom of the computational domain. In order to ensure the domain size is sufficiently large, we choose the height and the radius of the simulation domain to be 30× the heater size. In order to ensure the size and grid independence of our simulations, we perform simulations at a larger domain size (100×) and a finer mesh, and we find negligible change in the results (<0.1%). A Gaussian-shaped beam is selected for the heat source with a diameter similar to the measured value (7.6 μm). In order to verify our simulation results, a well-known material, quartz (c-SiO$_2$), is selected and its thermal conductivity is experimentally measured as a function of laser power. We then use our model to estimate the temperature gradient across the sample. It must be noted that the probe beam size is on the order of the heater beam size leading to a temperature gradient within the probed area on the sample. Therefore, the measured thermal conductivity is a weighted average of the thermal conductivity gradient in the probed volume in the quartz. To correctly account for this, we assume a cylindrical probed volume with radius and height similar to that of the probe and the

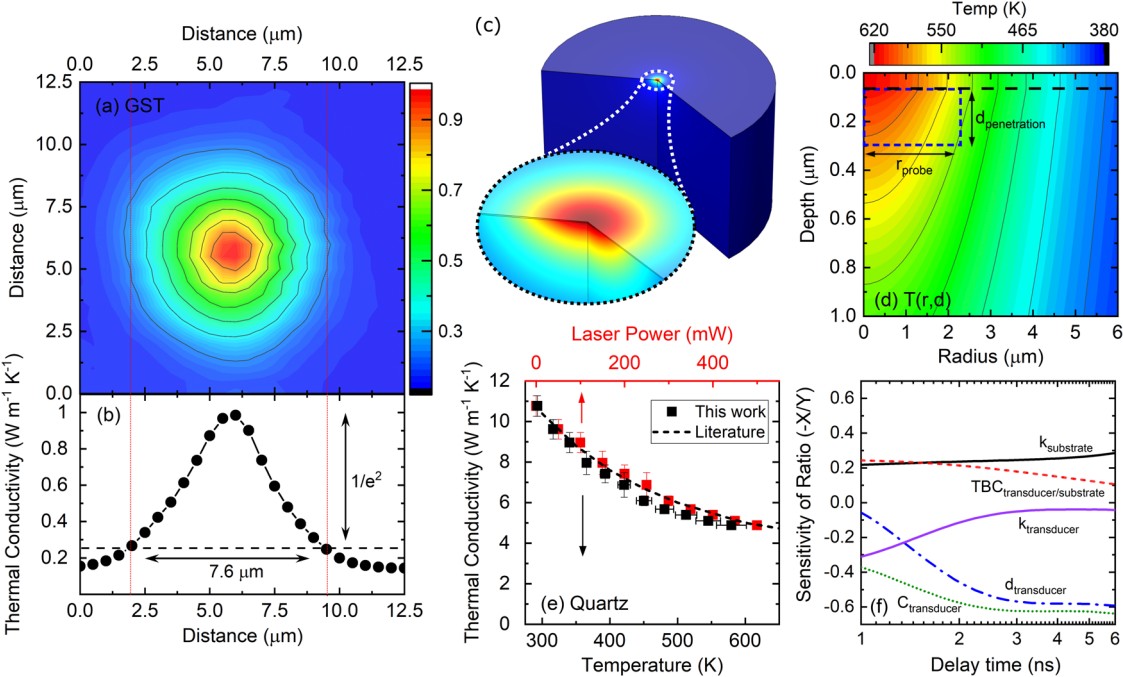

**Fig. 2 Thermal conductivity and temperature profile as a result of a localized heating source from a laser beam. a** The thermal conductivity map for a blanket coated sample with 60 nm ruthenium on a 40 nm thick phase-change material ($Ge_2Sb_2Te_4$) on a silicon substrate exposed to a heater laser with 740 mW. **b** Thermal conductivity profile of phase-change material as a function of distance. **c** Simulation domain for Al/Quartz and its corresponding temperature rise, (**d**) 2D temperature profile in the quartz and the region probed by TDTR beam, (**e**) thermal conductivity of quartz as a function of temperature and laser power. The x-axis and y-axis uncertainty are calculated based on standard deviations in temperature of probed volume and thermal conductivity measurement of three different spots, respectively. **f** Sensitivity analysis for the 80 nm SRO-transducer/60 nm PZO/15 nm SRO/DSO-substrate configuration for parameters such as thermal conductivity, $k$, volumetric heat capacity, $C$, thermal boundary conductance, TBC, and thickness, d.

thermal penetration depth of our probe beam ($d_p = \sqrt{k/\pi f C}$), with $k$ as the thermal conductivity, $f$ as the modulation frequency, and $C$ as the volumetric heat capacity. The probed region is demonstrated as the blue dashed rectangular in Fig. 2d. Using this approach, the thermal conductivity of quartz as a function of the mean temperature within the probed region is presented in Fig. 2e, which shows reasonable agreement with the literature.

**Reciprocal space mapping**. To confirm a phase transformation upon heating, we have performed reciprocal space map measurements of the epitaxial $PbZrO_3$ film from 300 to 523 K, as shown in Fig. 3. Example data collected around the $DyScO_3$ 332 reflection is shown below, as well as the $PbZrO_3$ $450_O$ reflection. The intensity of the $450_O$ reflection decreases with increasing temperature as the displacements of oxygen and lead ions from their lattice positions increase with temperature. It should be noted that the displacements, particularly of lead, have been shown to be much larger than typical thermal displacements and lead to local disorder[32]. The $450_O$ reflection is one that is not present in the cubic phase and is due to the large orthorhombic unit cell of the antiferroelectric phase. Likewise, the $PbZrO_3$ $440_O$ and $280_O$ orthorhombic reflections visible between 300 and 423 K merge into a single $013_C$ (where C denotes cubic) reflection between 473 and 523 K, which further confirms the transition to the higher symmetry cubic structure at elevated temperatures. The return of the $450_O$ reflection and splitting of the $440_O$ and $280_O$ reflections after cooling to room temperature emphasizes the reversibility of the AFE-to-PE phase transformation. We therefore show structural evolution from 300 to 523 K as $PbZrO_3$ reversibly transitions from the antiferroelectric to the paraelectric phase, which is consistent with our thermal conductivity measurements presented in the subsequent section.

The structural phase transition upon heating (AFE-to-PE) in PZO results in a higher symmetry crystal structure and should lead to less phonon-phonon scattering[33]. As a result, one would expect to observe an increase in the PZO thermal conductivity upon transitioning into the paraelectric phase. Although there is extensive research on the origin of the antiferroelectricity in PZO[34–36], the effect of field and thermal perturbation on its thermal conductivity is limited in the literature. Motivated by this, we are prompted to investigate how the electric field and temperature affect the PZO thermal conductivity. In the following, we will first demonstrate how the thermal conductivity of the PZO changes upon the application of an electric field. Then we present the temperature-dependent thermal conductivity data for epitaxial and polycrystalline samples using resistive heating. Once we show that the thermal conductivity of the PZO increases with temperature, we will show thermal conductivity dependence on the laser power. Finally, we will show how we can use electric field and optical heating to switch the thermal conductivity of PZO between various states.

**Thermal conductivity measurements**. Once we have established that the AFE-to-PE phase transition occurs in our sample and the heater laser is capable of sufficiently heating the film above its Curie temperature, the epitaxially grown PZO film is used to investigate the effects of electrical field and optical heating on its thermal conductivity. For this purpose, we adopt a widely used thermometry technique known as time-domain thermoreflectance (TDTR) that is capable of measuring the thermophysical properties of thin films with great accuracy (see Methods). In order to determine the thermal conductivity of epitaxial PZO from TDTR data, we assume a two-layer model (transducer-substrate), where we treat 60 nm PZO film as an interfacial layer. Figure 2f shows the

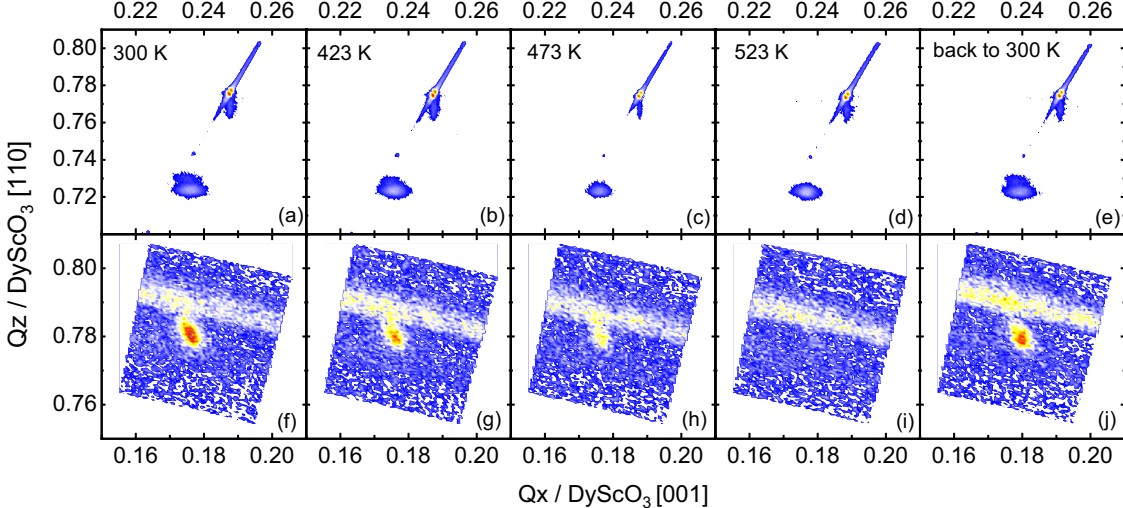

**Fig. 3 Reciprocal space maps at various temperatures to confirm antiferroelectric to paraelectric (AFE-to-PE) phase transition.** (**a**)–(**e**) show maps collected about the DyScO₃ 332 reflection and (**f**)–(**j**) show maps collected for the PbZrO₃ 450$_O$ reflection. The 450$_O$ reflection disappears between 473 and 523 K while the 440$_O$ and 280$_O$ reflections merge to a single peak above 423 °C. These data confirm that the antiferroelectric to paraelectric transformation in epitaxial PbZrO₃ occurs between 473 and 523 K. The streak denoted as the PEEK dome in (**f**)–(**j**) is a reflection from the protective dome on the hot stage and not part of the sample stack.

sensitivity of our measurements to various parameters, considering a two-layer model. According to this, the TDTR measurements are highly sensitive to the volumetric heat capacity and the thickness of the transducer. We obtain the volumetric heat capacity of the transducer from the literature and calculate the uncertainty of our measurements based on the transducer thickness. To increase the sensitivity of our measurements to the intrinsic thermal conductivity of the film, we also measure 300 nm polycrystalline PZO as a function of temperature. For consistency in the analyses of the results between field and thermal effects, we assume constant values for heat capacity and transducer thickness and fit for the thermal boundary conductance (TBC$_{transducer/substrate}$) and substrate thermal conductivity ($k_{substrate}$). Note, since the volumetric heat capacities of both the SRO transducer and DSO substrate increase with temperature[37,38], this method likely leads to a slight under-prediction of the increase observed in thermal conductivity upon heating.

In order to determine the effects of electrical bias on thermal conductivity, an electric field is applied across 60 nm of PZO deposited on a DyScO₃ (DSO) substrate and its thermal conductivity is measured as a function of voltage. For applying an electric field in the cross-plane direction across the PZO, the film is deposited on 15 nm of SrRuO₃ which serves as the bottom electrode, and subsequent to PZO deposition, circular electrical contacts (80 nm of SrRuO₃) are deposited on top of the PZO layer to serve as the top electrode (see the inset in Fig. 4c). These results are presented in Fig. 4a showing that the thermal conductivity drops by nearly 10% upon applying a positive/negative electric field and immediately returns to its original state after removing the electric field. This is consistent with what one would expect for an antiferroelectric material which returns to its original non-polar state upon the elimination of electric field. In order to provide some insight into the observed changes in thermal conductivity as a function of electric field, we use a Callaway-type model and provide an estimation for thermal conductivity by taking multiple scattering parameters into account. According to these calculations, for the thermal conductivity of PZO to be reduced by ~10% upon application of electric field, the domain size must shrink by 18% and 20%, for domains with 3–30 nm width, respectively (see Supplementary Note 1).

As discussed earlier, this switching behavior in thermal conductivity has been observed in PbZr₀.₃Ti₀.₇O₃/ PbZr₀.₇Ti₀.₃O₃ ferroelectric bilayers and was attributed to the formation of new ferroelastic domains under applied electric field, leading to increases in the domain-boundary density[9]. The changes in the domain-boundary populations are on the order of hundreds of nanoseconds and enable ultrafast switching of these materials[39]. The increased boundary area between the domains results in greater phonon-boundary scattering leading to a reduction in thermal conductivity upon the application of an electric field. As shown in Fig. 1b, with the application of fields greater than ~380 kV cm⁻¹, the AFE phase of the epitaxial film transforms into a ferroelectric phase. The same field required for that transformation is the onset of the decrease in thermal conductivity, which suggests that the decrease is related to the phase transition. We postulate that upon transitioning to the ferroelectric phase, new ferroelastic domains form with an accompanying increase in ferroelastic domain-boundary area, which increases the phonon scattering rates and decreases thermal conductivity. This is consistent with the decrease observed in thermal conductivity of PZO upon application of an electric field similar to that of PZT and may be expected because the film is clamped to a mechanically rigid substrate.

Figure 4b shows the thermal conductivity of the epitaxial and polycrystalline PZO samples as a function of temperature. For these measurements, we assume constant properties at room temperature with a specific heat of 2.52 MJ m⁻³ K⁻¹ for polycrystalline PZO, taken from ref. [40]. The thermal conductivity of PZO follows a glass-like behavior at low temperatures and plateaus above 200 K. Above room temperature, the thermal conductivity gradually increases with temperature up to ~500 K where the PZO transitions from an AFE-to-PE phase leading to an increase in thermal conductivity from ~1.30 to ~1.65 W m⁻¹ K⁻¹. This trend is observed across multiple samples with different microstructures, thicknesses, and transducers as presented in Fig. 4b. The smoothly increasing thermal conductivity is unusual for a highly crystalline material and shows that the complex octahedral tilts and lead-ion displacements in PbZrO₃ lead to high phonon scattering rates. Upon transitioning to the cubic phase, the material becomes globally cubic; the octahedral tilts cease to

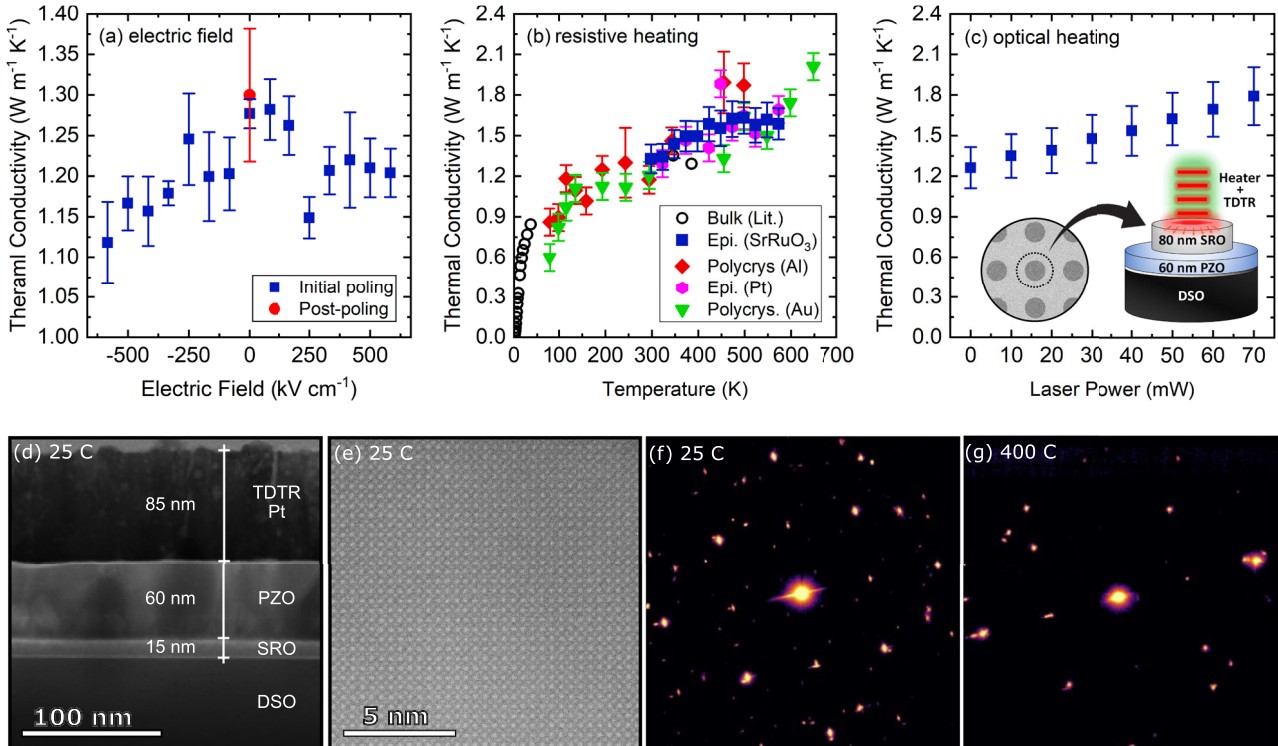

**Fig. 4 Thermal conductivity of PZO upon phase transformation as a result of electric field and thermal stimuli. a** Thermal conductivity as a function of electric field for PZO measured at room temperature. **b** Thermal conductivity as a function of temperature, measured on a resistive heating stage. The uncertainty in (**a**, **b**) is based on standard deviation across multiple scans. **c** Thermal conductivity as a function of laser power. The uncertainty in (**c**) is calculated based on 10% variations in the transducer thickness, and the inset shows the schematic of the layers configuration studied here. **d** Annular bright-filed STEM image showing the corresponding layers in the epitaxial PZO at room temperature, (**e**) and the high-resolution TEM showing the high degree of order in the studied sample. Selected-area diffraction patterns of epitaxial PZO at (**f**) 25 °C and (**g**) 400 °C showing that our PZO film has undergone a phase transition from orthorombic space group *Pbam* to cubic *Pm3̄m* (see Supplementary Note 2).

exist. This cubic symmetry leads to an increase in thermal conductivity. The increase in thermal conductivity, however, is likely smaller than we would anticipate for such a dramatic change in crystal symmetry (i.e. an 8X reduction in unit cell volume). The existence of local lead-ion disorder, which occurs on the length scales of phonon wavelengths, impacts the overall achievable thermal conductivity in the cubic phase.

This increase in thermal conductivity is related to the structural transition to a higher symmetry crystal structure in the PZO. The complex orthorhombic unit cell of PZO at room temperature transforms into a cubic unit cell above the phase-transition temperature. This transition into a simpler, higher symmetry cubic crystal structure reduces the number of atoms (N) in the conventional unit cell and, as a result, leads to less phonon scattering[33]. We confirm the phase transition for both epitaxial and polycrystalline films using scanning transmission electron microscopy (STEM). The lower magnification image shown in Fig. 4d shows the layers in the epitaxial PZO sample. High-resolution Z-contrast images and selected-area diffraction patterns (SADP) (Figs. 4e, f, respectively) show that the grains are oriented with pseudo-cubic axes out-of-plane. Ordered reflections occurring in the room temperature SADP that indicate the presence of octahedral rotations disappear when the sample is heated to 400 °C, as shown in Figs. 4f, g with further annotation in Supplementary Note 2. The disappearance of the ordered reflections confirms the AFE-to-PE phase transition. In contrast to the AFE-to-PE phase transition observed here, our thermal conductivity measurements on bulk commercial PZO using the transient plane source technique show no change in thermal conductivity above room temperature. As we show

in Supplementary Note 3, this is due to the existence of porosity, disorder, and impurity in the bulk PZO sample.

We repeat high-temperature measurements using a focused laser source to locally heat the PZO and measure the thermal properties within the laser-heated region. For this experiment, we use epitaxial 60 nm PZO and perform the experiment on the SRO contacts. Unlike resistive stage heating where the entire sample is raised to the temperature of interest, laser heating creates a temperature gradient across the sample where the maximum temperature is on the surface and in the middle of the focused laser spot with a Gaussian intensity profile. The existence of a temperature gradient in the heated area is closer to an actual device configuration and allows us to capture a more realistic change in thermal conductivity. Given that, after measuring the thermal conductivity as a function of laser power, similar to resistive heating, we observe a gradual increase in thermal conductivity with laser power. This gradual increase is consistent with the static temperature measurements where the continuously increasing thermal conductivity with temperature was observed.

Thus far, we have demonstrated that the thermal conductivity of PZO decreases upon exposure to an electric field and increases upon raising the temperature to the Curie temperature. Now, we demonstrate how these field and thermal effects can be used in tandem to create a bidirectional thermal conductivity switch. For this, we periodically apply electrical and optical excitation to the 60-nm-thick epitaxial PZO sample and switch the material between low- and high-thermal conductivity states. Figure 5a, b show the switching mechanism in real-time when the PZO is under periodic electric field and heating, respectively. For an electric field amplitude of 670 kV cm$^{-1}$ the thermal conductivity

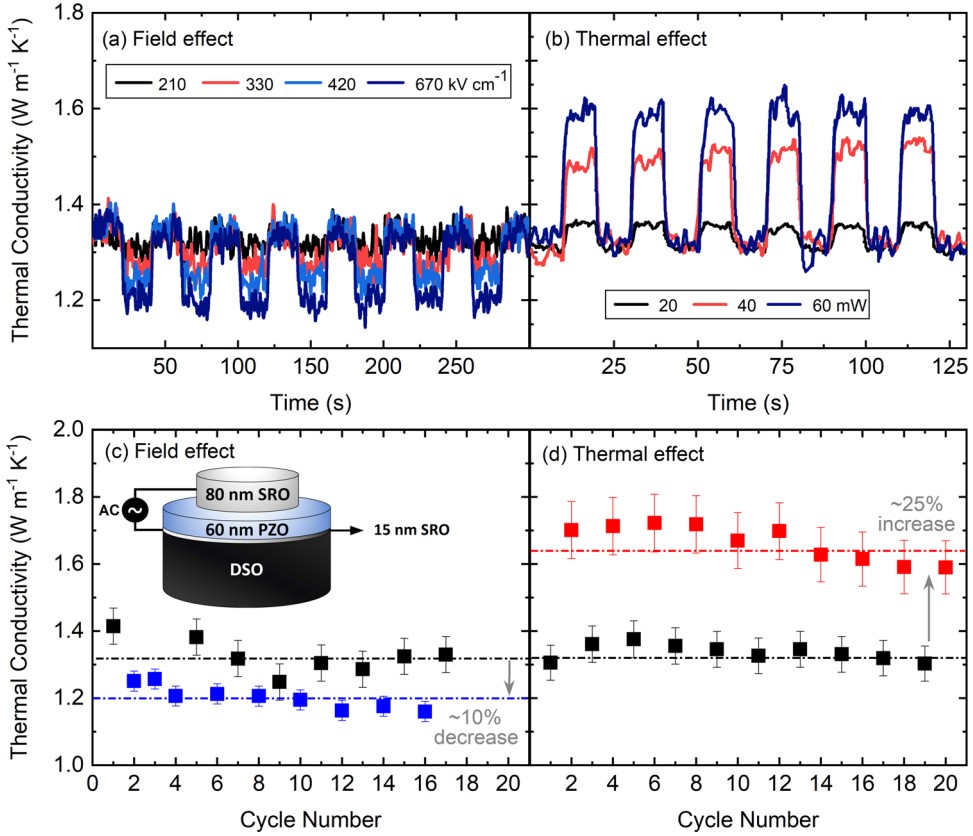

**Fig. 5 Real-time switching of epitaxial PZO to low and high thermal conductivity using electrical and thermal stimuli.** Switching thermal conductivity of PZO as a function of time measured at 500 ps delay time for (**a**) electric fields of 210, 330, 420, and 670 kV cm$^{-1}$, and (b) heater laser powers of 20, 40, and 60 mW. **c**, **d** Repeatability of switching thermal conductivity upon applying maximum electric field and laser power before damaging the sample. The dashed lines represent the weighted average of the data points corresponding to that line. These measurements were performed at a single location on the sample. The uncertainty is calculated based on a 10% change in transducer thickness.

can be periodically decreased by nearly 10%. On the other hand, upon applying optical heating using a laser with a spot size of 12 $\mu$m in diameter, the PZO thermal conductivity increases by nearly 25%. Figure 5c, d show this transition is repeatable for a number of cycles. Although switching thermal conductivity using electric field shows some drift from one cycle to another, we attribute this to the existence of a small sensitivity to the film thermal conductivity rather than inherent changes in the film due to cycling or due to different domain structures upon each field cycle. On the other hand, we observe a small decay in thermal conductivity of PZO as well as reflectivity signal after multiple switching cycles with the laser source, which is attributed to gradual degradation in the SRO transducer due to heating cycles. In Supplementary Note 4 we provide an estimated temperature rise on the surface of SRO using previously developed models.

In summary, we demonstrated the thermal conductivity of antiferroelectric PZO can be bidirectionally switched by −10% and +25% upon application of electric field and thermal excitation, respectively. Similar to ferroelectric materials where the application of electric field increases the domain-wall density and reduces the thermal conductivity, we observe that the thermal conductivity of antiferroelectric PZO decreases upon applying an electrical bias across the sample. Furthermore, we take advantage of PZO's relatively low Curie temperature (220 °C) and, using optical heating, we induce the orthorhombic to cubic phase transition, resulting in higher crystal symmetry and increased thermal conductivity in PZO. According to our results, the net thermal conductivity switching ratio that can be obtained in PZO is around 38%.

## Methods

**Chemical solution deposition (CSD).** Polycrystalline PbZrO$_3$ films were prepared by chemical solution deposition using an inverted mixing order chemistry[41]. The solution comprised zirconium butoxide (80 wt.% in butanol, Sigma Aldrich), lead (IV) acetate (Sigma Aldrich), methanol, and acetic acid. A 0.35 M solution was used with 35 mol% excess Pb to account for loss to the substrate and atmosphere during high-temperature annealing. The PZO film was coated on a 100 nm Pt/40 nm ZnO/300 nm SiO$_2$/(001) Si substrate by spin casting at 4000 RPM for 30 s followed by a pyrolysis step at 350 °C for 1 min[42]. After four coating and pyrolysis steps, an additional layer of a 0.1 M PbO solution was coated onto the surface to provide a PbO overpressure and the film was annealed at 700 °C for 10 min in a preheated box furnace[43]. The final PbZrO$_3$ film thickness was 300 nm. X-ray diffraction was performed using a PanAlytical Empyrean diffractometer in a Bragg–Brentano geometry with a GaliPIX detector and Cu Kα radiation, and scanning electron microscopy was performed using a FEI Quanta 650 with a concentric backscatter detector and an acceleration voltage of 15 kV[44].

**Time-domain thermoreflectance (TDTR).** The thermal conductivity of the PZO samples was measured using time-domain thermoreflectance (TDTR). The details of the measurement technique and thermal model that relates the experimental data to thermal properties are given elsewhere[45–49]. In short, and related to the current manuscript, the output of a pulsed Ti:sapphire laser (80 MHz, 14 nm FWHM, and 808 nm center wavelength) is split into a pump and a probe path where the pump path is electro-optically modulated at a frequency of 8.4 MHz to create an oscillatory heating event on the surface of the sample. The probe path is directed to a mechanical delay stage to capture the temporal change in the thermoreflectivity of the sample which can be related to the changes in temperature. In order to facilitate the detection of changes in thermoreflectivity of the surface, we deposit 80 nm of metallic transducer on top of the sample prior to taking the measurements. For thermal conductivity measurements at elevated temperatures, we use (i) resistive heating with a commercial temperature control stage (Linkam model HFS600E-PB4) and (ii) optical heating with a CW laser operating at 532 nm wavelength. The temperature control stage enables uniformly heating the entire sample to a known temperature, while the laser heating allows for rapid (sub-second) temperature rise by hundreds of degrees on the sample's surface.

**Laser heating**. In order to optically heat the sample, we use a CW laser and, by passing it through an electro-optic modulator (EOM), we modulate the heater beam at the same frequency as the TDTR pump beam. The EOM on the heater's path is used for overlapping the heater with TDTR beams by maximizing the magnitude signal. All the beams pass through the same ×20 objective and using a sapphire calibration sample, we measure the size of the TDTR beams by fitting for spot sizes. Using this approach, we estimate the pump and probe diameters to be 5 $\mu$m and 5 $\mu$m, respectively. To ensure the accuracy of the results, we use the obtained spot sizes and fit for the thermal conductivity of a-$SiO_2$ and quartz. We find excellent agreement between the measured thermal conductivity and the literature.

**X-ray diffraction**. Temperature-dependent reciprocal space maps were collected on a Rigaku Smartlab diffractometer with Cu K$\alpha$ radiation using a parallel beam geometry under air atmosphere. An Anton Paar DHS 900 hot stage with a PEEK (polyetheretherketone) dome was used to heat the sample for each measurement.

**Scanning transmission electron microscopy**. (S)TEM experiments of the epitaxial PZO films were performed on a Thermo Fisher Scientific Themis Z-STEM operating at 200 kV. In STEM mode the probe current was limited to less than 50 pA. TEM SADP images were acquired using the Ultra Scan camera. TEM experiments on the CSD PZO film were performed on a FEI Titan operating at 300 kV equipped with a Gatan OneView-IS camera. All heating experiments used a Gatan heating holder.

**Reporting summary**. Further information on research design is available in the Nature Research Reporting Summary linked to this article.

## Data availability
The data that support the findings of this study are available from the corresponding author upon reasonable request.

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

## Acknowledgements

R.G. acknowledges support from the National Science Foundation under Grant DMR-1708615. L.W.M. acknowledges support from the Army Research Office under Grant W911NF-21-1-0118. A.S., E.R.H., E.J.O., J.F.I., and P.E.H. acknowledge support from a Research Innovation Award from the University of Virginia. K.A., M.S.B.H., and J.F.I. acknowledge support from the National Science Foundation under Grant DMR-2006231. S.M. was supported by the University of Virginia School of Engineering and Applied Sciences to join the Biotechnology Training Program (an NIH training grant). This work was supported in part by the NSF I/UCRC on Multi-functional Integrated System Technology (MIST) Center IIP-1439644, IIP-1439680, and IIP-1738752. This material is based upon work supported by the U.S. Department of Energy's Office of Energy Efficiency and Renewable Energy (EERE) under the Building Technologies Office (BTO) award number DE-EE0009157. This report was prepared as an account of work sponsored by an agency of the United States Government. Neither the United States Government nor any agency thereof, nor any of its employees, makes any warranty, express or implied, or assumes any legal liability or responsibility for the accuracy, completeness, or usefulness of any information, apparatus, product, or process disclosed, or represents that its use would not infringe privately owned rights. Reference herein to any specific commercial product, process, or service by trade name, trademark, manufacturer, or otherwise does not necessarily constitute or imply its endorsement, recommendation, or favoring by the United States Government or any agency thereof. The views and opinions of authors expressed herein do not necessarily state or reflect those of the United States Government or any agency thereof.

## Author contributions

K.A., J.A.T., and P.E.H. designed the experiments. R.G., A.S., J.N., J.C.R., L.W.M., and J.F.I. made the samples. K.A., J.A.T., M.S.B.H., S.M., T.P., and J.L.B. performed the thermal conductivity measurements as a function of temperature. J.A.T., D.H.O., performed thermal conductivity measurements as a function of electrical bias. K.A. performed the simulations. R.G., E.R.H., T.M., A.S, L.W.M., and J.F.I. performed the characterizations. K.A., J.A.T., L.W.M., J.F.I., and P.E.H wrote the manuscript.

## Competing interests

The authors declare no competing interests.
