## [Peer review file · Nature Communications]

Reviewers' comments:

Reviewer #1 (Remarks to the Author):

This work focused on the solid-state bidirectional thermal conductivity switching in antiferroelectric lead zirconate which has been rarely studied. With the help of heating or/and electric field, the phase transition in the material could lead to a volume expansion or/and reduction in the unit-cell size, and also possibility of altering the populations of ferroelastic domains, finally cause a change of thermal conductivity. Although the authors successfully demonstrated that the thermal conductivity truly could be tuned by electric field or/and heating, the explanations are not so solid. Some results in the manuscript are suggested to be further explained or reinterpretation, especially the reason about the change of the thermal conductivity should be somewhat quantitatively given. For example, the authors claimed that "The increased boundary area between the domains results in a greater phonon-boundary scattering leading to a reduction in thermal conductivity upon the application of electric field". Actually, some well-known model could be used to calculate the boundary-caused reduction of thermal conductivity, which should be compared to the experiments for confirming the explanation. The effects of volume expansion or/and reduction in the unit-cell size should be also quantitatively confirmed. In my opinion, this paper can be accepted after modification.

Reviewer #2 (Remarks to the Author):

This manuscript describes the bidirectional tuning of the thermal conductivity of the PbZrO₃ (PZO) film by electric field and laser irradiation. Epitaxial PZO film was prepared by pulsed laser deposition, and the thermal conductivity of the film was measured using the TDTR technique. Thermal conductivity of the film was decreased 10 % by the applied electric fields and increased 25 % by the laser heating. Sample deposition and the TDTR measurement methods were described in the literature. The laser heating method was described in this manuscript.

The author claimed that the PZO film can serve as the thermal switch, however, the laser heating more than 100 °C is obviously inappropriate for such kind of device. If the thermal baths were bridged by the device, and the device was heated up from RT to 500 K, a device might be no longer a heat conductor but a heat spreader.

Temperature dependence of the thermal conductivity of a PZO film according to crystal symmetry change is an attracting topic as well as field induced domain boundary modulation. Furthermore, realtime thermal conductivity mapping equipped with a pulsed heating laser is also of interest of researchers of condensed matters. However, from the viewpoint of the development of thermal switching device, this manuscript is not recommended publishing in Nature Communications.

Reviewer #3 (Remarks to the Author):

I have reviewed this manuscript and find that it is not suitable for publication at the present stage. My main concern about this work is that the effect of the substrates on the thermal conductivity changes upon the electric field and thermal excitation in the authors' measurement is not provided to prove that this bidirectional thermal conductivity switching behavior is the intrinsic property from PZO.

Although this switching behavior can be clearly demonstrated for PZO film, the bulk PZO sample does not exhibit dramatic thermal conductivity changes upon the transition region, as the authors showed in the supplementary information. The authors attributed it to existence of porosity and disorder or impurity in the bulk sample, but the measurement for a pure sample should be performed. On the other hand, the thermal conductivity measurement and the adjustment upon the electric field and thermal excitation should be performed on the substrates, and the results should be compared with that of the PZO film. These measurements would be a strong evidence for confirming this observation. My other comments I made while reading through this manuscript are listed below:

1. The temperature unit should be unified to either degree C or Kelvin.

2. What is the measurement uncertainty for the film thermal conductivity? How was the error bars calculated in figure 3.
3. In figure 3(b), the thermal conductivity data in the transition should be presented in a inset to clearly see the thermal conductivity jump in this region.
4. In figure 4(b), the thermal conductivity remains almost no change upon 20 mW heating power, indicating the sample's temperature is below the transition region, while for the increase of thermal conductivity with heating power of 40 and 60 mW, the sample's temperature should be over the transition temperature. What is the approximate temperature of the sample in the heating power of 20, 40 and 60 mW?
5. The switching thermal conductivity behavior observed in the PZO film reported in the work should also be compared with that of other related ferroelectric materials reported in literature.

Reviewer #1 (Remarks to the Author):

This work focused on the solid-state bidirectional thermal conductivity switching in antiferroelectric lead zirconate which has been rarely studied. With the help of heating or/and electric field, the phase transition in the material could lead to a volume expansion or/and reduction in the unit-cell size, and also possibility of altering the populations of ferroelastic domains, finally cause a change of thermal conductivity. Although the authors successfully demonstrated that the thermal conductivity truly could be tuned by electric field or/and heating, the explanations are not so solid. Some results in the manuscript are suggested to be further explained or reinterpretation, especially the reason about the change of the thermal conductivity should be somewhat quantitatively given. For example, the authors claimed that “The increased boundary area between the domains results in a greater phonon-boundary scattering leading to a reduction in thermal conductivity upon the application of electric field”. Actually, some well-known model could be used to calculate the boundary-caused reduction of thermal conductivity, which should be compared to the experiments for confirming the explanation. The effects of volume expansion or/and reduction in the unit-cell size should be also quantitatively confirmed. In my opinion, this paper can be accepted after modification.

Authors Response: We would like to thank the reviewer for taking the time and reviewing our manuscript. **Following the reviewer’s suggestion about comparing our experimental results against theoretical models, we turned to a Callaway-type model [1] that is widely used to estimate the thermal conductivity of crystalline materials [2-7].** This model takes various scattering parameters into account such as Umklapp, defect, and boundary scattering:

$$k = \frac{k_B}{2\pi^2v} \int_0^{k_B\Theta_D/\hbar} \tau \frac{\hbar^2\omega^2}{k_B^2T^2} \frac{e^{\hbar\omega/k_B T}}{(e^{\hbar\omega/k_B T} - 1)^2} \omega^2 d\omega$$

where k_B and \hbar are the Boltzmann’s and Planck’s constants, respectively, v is the speed of sound, Θ_D is the Debye temperature, τ is relaxation time, ω is the vibrational modes frequency, and T is temperature. We can estimate the relaxation time from different scattering mechanism via Matthiessen’s rule:

$$\tau^{-1} = A\omega^4 + B\omega^2Te^{-C/T} + v/d$$

where d is the scattering length scale, and A , B , and C , are the scattering coefficients for impurity scattering ($A\omega^4$) and Umklapp scattering ($B\omega^2Te^{-C/T}$). Pertinent to this study, since the domain wall density increases upon electrical biasing, we change the *scattering length scale* (d) to match our thermal conductivity measurements. For this, we fix scattering length scale to the domain size in PZO and fit A , B , and C to our experimental data as a function of temperature as shown in Fig. 1. According to our previous work [8], the antiferroelectric domains in our PZO film are oriented at 90° and display correlation lengths on the order of **3 and 30 nm** for the $[\bar{1}01]$ and $[101]$ orientations, respectively. Assuming $d = 3$ and 30 nm, we fit for A , B , and C . Once, these scattering coefficients are determined, we calculate how much the scattering length scale, d , must change upon electrical biasing to match our experimental observations. **According to these calculations, in order to reduce the thermal conductivity of PZO by ~10% for 3 and 30 nm domain size, d must change by 18% and 20%, respectively.** This degree of change in scattering length scale is comparable to previously reported values (~10%) regarding percentage of change in the domain wall density for ferroelectric materials [9]. This discussion is added to the Supporting information.

Table 1. PZO thermophysical properties used in our Callaway model [6].

Unit cell size (nm)	Longitudinal Sound speed, v_{LA} (m/s)	Transverse Sound speed, v_{TA} (m/s)	Atomic density (n) (nm^{-3})	Debye temperature, Θ_D (K)
0.417	4928	2746	69.36	391

Figure 1. Measured thermal conductivity of PZO as a function of temperature. The dashed line shows the Callaway-type model's fit for the domain size (d) of 3 nm.

Additionally, in order to quantitatively investigate the effect of volume change upon temperature, we used our selected area diffraction pattern from TEM at room temperature and 400 °C. According to these measurements, the distance, g , from the forward scattered (000) peak to the $\{002\}_{pc}$ Bragg peaks are measured from the selected area diffraction patterns shown in Fig. 2. The lattice parameters, a , are then calculated as $a = 2/g$. The room temperature pseudocubic in-plane and out-of-plane lattice parameters are estimated to be 4.143 ± 0.01 and 4.177 ± 0.01 Å, respectively. At 400°C the in-plane and out-of-plane lattice parameters are 4.153 ± 0.004 and 4.179 ± 0.02 Å, respectively. The uncertainties are the distance from the mean values to either g . The ± 0.01 Å uncertainty amounts to a $\sim 0.2\%$ change from the reported means. Using the definition for the linear coefficient of thermal expansion, $\alpha = \Delta a / (a_0 \Delta T)$, we estimate it to be 6.3×10^{-6} , which is comparable to the reported value of 9.2×10^{-6} [6]. Alternatively, using the reported value for coefficient of thermal expansion from Ref. [6] to calculate $\Delta a / a_0$ we find an expansion of 0.2%, which is at the limit of our 0.2% measurement uncertainty. **Therefore, using the selected area diffraction patterns presented in Fig. 2, we measured a linear thermal expansion coefficient agreeable with literature, which is at the bound of our experimental error.**

Figure 2. SAED of $\langle 110 \rangle_{pc}$ zone axis grains at (a) 25 °C and (b) 380 °C. At 25 °C $\frac{1}{2} \langle 111 \rangle_{pc}$ ordered reflections are marked with red arrow and indicate the presence of the orthorhombic anti-ferroelectric phase. After heating no ordered reflections were observed in any $\langle 110 \rangle_{pc}$ zone axis, as seen in (b).

Furthermore, we have performed additional reciprocal space map measurements of the epitaxial PbZrO_3 film. We have performed measurements at temperatures spanning between 25 and 250 °C. Example data collected around the DyScO_3 332 reflection is shown below as well as the PbZrO_3 450_o (where o denotes the orthorhombic phase) reflection. The PbZrO_3 reflection clearly shows that the orthorhombic phase disappears at a temperature between 200 and 250 °C and reappears after cooling to room temperature. The 450_o reflection is one that is not present in the cubic phase and results due large orthorhombic unit cell of the antiferroelectric phase. Likewise, the PbZrO_3 440_o and 280_o orthorhombic reflections are clearly visible between 25 and 150 °C, but have merged to a single 013_c (where c denotes cubic) reflection between 200 and 250 °C. Peak splitting in the 440_o and 280_o reflections is again present after cooling to room temperature, showing reversibility of the antiferroelectric/paraelectric phase transition. **These data clearly show that the orthorhombic phase reduces in unit cell size upon heating above the antiferroelectric to paraelectric phase transition temperature.**

Figure 3. X-ray diffraction reciprocal space maps collected over a temperature range of 25 to 250 °C for epitaxial PbZrO_3 on DyScO_3 . (a)-(e) show maps collected about the DyScO_3 332 reflection and (f)-(j) show maps collected for the PbZrO_3 450_o reflection. The 450_o reflection disappears between 200 and 250 °C while the 440_o and 280_o reflections merge to a single peak above 150 °C. These data confirm that the antiferroelectric to paraelectric transformation in epitaxial PbZrO_3 occurs between 200 and 250 °C. The streak denoted as the PEEK dome in (f)-(j) is a reflection from the protective dome on the hot stage and not part of the sample stack.

Regarding the electric field induced phase transition and volume expansion, we are unable to experimentally measure that expansion with the samples in this study. **The polycrystalline thin film cannot withstand the electric field for a sufficiently long period of time to allow for an in-situ X-ray diffraction experiment and the electrodes on the epitaxial film are too small for a lab-scale X-ray diffractometer to focus on a single pad to actuate and demonstrate the phase transition and volume expansion.** Synchrotron X-ray microdiffraction would be necessary to make such an observation, for which we do not have access. While we cannot experimentally validate this effect directly, we can direct the reviewer and interested readers to the literature where this has been confirmed. For example, recent in-situ synchrotron X-ray diffraction work by Liu, et al. has shown that upon applying an electric field sufficient to transform an antiferroelectric PbZrO₃-base ceramic to the *R3c* ferroelectric phase, a lattice strain of 0.3% develops for the 110_p and 111_p reflections [10]. This translates to a volume strain of 0.4%. Other work by Frederick, et al. measured this strain macroscopically (i.e. averaged over the whole volume of a polycrystalline solid) and also showed a 0.4% strain upon transitioning to the ferroelectric phase [11], which has also been measured by macroscopic means. These positive strain values are volume expansions upon transitioning from the antiferroelectric to ferroelectric phase.

Reference

- [1] Callaway, Joseph. "Model for lattice thermal conductivity at low temperatures." *Physical Review* 113.4 (1959): 1046.
- [2] Thacher, Philip D. "Effect of boundaries and isotopes on the thermal conductivity of LiF." *Physical Review* 156.3 (1967): 975.
- [3] Wei, Lanhua, et al. "Thermal conductivity of isotopically modified single crystal diamond." *Physical review letters* 70.24 (1993): 3764.
- [4] Olson, J. R., et al. "Thermal conductivity of diamond between 170 and 1200 K and the isotope effect." *Physical Review B* 47.22 (1993): 14850.
- [5] Mingo, Natalio, et al. "Predicting the thermal conductivity of Si and Ge nanowires." *Nano Letters* 3.12 (2003): 1713-1716.
- [6] Foley, B. M. et al. Phonon scattering mechanisms dictating the thermal conductivity of lead zirconate titanate (PbZr_{1-x}Ti_xO₃) thin films across the compositional phase diagram. *Journal of Applied Physics* 121, 205104 (2017).
- [7] Scott, Ethan A., et al. "Reductions in the thermal conductivity of irradiated silicon governed by displacement damage." *Physical Review B* 104.13 (2021): 134306.
- [8] Gao, Ran, et al. "Ferroelectricity in Pb_{1+δ}ZrO₃ thin films." *Chemistry of Materials* 29.15 (2017): 6544-6551.
- [9] Ihlefeld, Jon F., et al. "Room-temperature voltage tunable phonon thermal conductivity via reconfigurable interfaces in ferroelectric thin films." *Nano letters* 15.3 (2015): 1791-1795.
- [10] Liu, H., L. Fan, S. Sun, K. Lin, Y. Ren, X. Tan, X. Xing, and J. Chen, "Electric-field-induced structure and domain texture evolution in PbZrO₃-based antiferroelectric by in-situ high-energy synchrotron X-ray diffraction," *Acta Materialia*, 184 41-49, (2020).
- [11] Frederick, J., X. Tan, and W. Jo, "Strains and Polarization During Antiferroelectric-Ferroelectric Phase Switching in Pb_{0.99}Nb_{0.02}[(Zr_{0.57}Sn_{0.43})_{1-y}Tiy]_{0.98}O₃ Ceramics," *Journal of the American Ceramic Society*, 94(4), 1149-1155, (2011).

Reviewer #2 (Remarks to the Author):

This manuscript describes the bidirectional tuning of the thermal conductivity of the PbZrO₃ (PZO) film by electric field and laser irradiation. Epitaxial PZO film was prepared by pulsed laser deposition, and the thermal conductivity of the film was measured using the TDTR technique. Thermal conductivity of the film was decreased 10 % by the applied electric fields and increased 25 % by the laser heating. Sample deposition and the TDTR measurement methods were described in the literature. The laser heating method was described in this manuscript.

The author claimed that the PZO film can serve as the thermal switch, however, the laser heating more than 100 °C is obviously inappropriate for such kind of device. If the thermal baths were bridged by the device, and the device was heated up from RT to 500 K, a device might be no longer a heat conductor but a heat spreader.

Temperature dependence of the thermal conductivity of a PZO film according to crystal symmetry change is an attracting topic as well as field induced domain boundary modulation. Furthermore, realtime thermal conductivity mapping equipped with a pulsed heating laser is also of interest of researchers of condensed matters. However, from the viewpoint of the development of thermal switching device, this manuscript is not recommended publishing in Nature Communications.

Authors Response: We would like to thank the reviewer for taking the time and reviewing our manuscript. In regards to the concerns raised by the respectable reviewer for the practicality of PZO as a thermal switch, we would like to highlight a few aspects of our work to elucidate its impact for the broad readership of *Nature Communications* and address the reviewer's concerns.

To the best of our knowledge, our work is the first experimental demonstration of a bi-directional solid-state thermal conductivity switch with no moving parts. Although there are several other works that reported on bi-directional thermal conductivity switching [1-3], all of these studies are either using liquid/gap-based switching or are not experimental. Our work not only provides physical insight into the mechanism of switching in PbZrO₃ which is very important from a fundamental condensed matter physics perspective but also demonstrates the implementation of PZO as a solid-state bi-directional thermal switch. In this sense, we believe our approach for switching the thermal conductivity of an insulating material that is **purely driven by phonons** is novel and will provide valuable insight for future studies regarding thermal management.

With regards to the reviewer's point about impracticality of temperatures above 100 °C from a device perspective, we need mention that **tuning thermal conductivity finds more applications in the fields outside of wearable electronics such as of space explorations [4-7], automotive [8-10], energy storage [11-14], and thermal management [15-17], all of which are concerned with temperatures close or above 500 K.** For instance, in deep space exploration where electronic devices are prone to damage due to intensive radiations or extreme temperatures, thermal switches are an integral component in the development of thermal logic devices that use heat instead of electricity to perform computation [18,19]. Alternatively, in motor vehicles, the sensors in the combustion chamber or exhaust system operate well above 500 K [9]. At the same time, from a pure fundamental physical perspective, our work would find great interest in the field of thermal rectifier/diodes [20-26].

Regarding the point raised by the respectable reviewer about the thermal bridge at 500 K, if we understand the reviewer correctly (i.e. heat conductor \approx heat spreader), **depending on the design, application, and purpose of using a thermal switch for a specific device, the spontaneous switching of PZO to higher thermal conductivity at 220 °C would be beneficial.** For instance,

in many applications when the temperature is increased, thermal switches are used to conduct heat with better efficiency by switching to higher thermal conductivity. Similarly, in this application, PZO can help as a spontaneous heat spreader at elevated temperatures with no need for a trigger mechanism to switch the PZO to high thermal conductivity state.

Nonetheless, **the focus of this work is not on developing an industrial-level thermal conductivity switch that can be readily integrated into devices.** Our work provides a fundamental understanding about how a bi-directional thermal conductivity switch can be achieved and paves the way for future studies to take advantage of multiple phase change mechanisms in materials for tuning the thermal transport in materials. For instance, there are other anti-ferroelectric materials with lower Curie temperatures such as (Pb,Ba)ZrO₃ (~110 °C) that could possibly be more relevant for the applications the reviewer is suggesting [27].

References

- [1] Liu, Chenhan, et al. "Bidirectional Tuning of Thermal Conductivity in Ferroelectric Materials Using E-Controlled Hysteresis Characteristic Property." *The Journal of Physical Chemistry C* 124.48 (2020): 26144-26152.
- [2] Lu, Qiyang, et al. "Bi-directional tuning of thermal transport in SrCoO_x with electrochemically induced phase transitions." *Nature materials* 19.6 (2020): 655-662.
- [3] Chae, Uikyu, et al. "A Hybrid RF MEMS Switch Actuated by the Combination of Bidirectional Thermal Actuations and Electrostatic Holding." *IEEE Transactions on Microwave Theory and Techniques* 68.8 (2020): 3461-3470.
- [4] Phoenix, Austin A., and Evan Wilson. "Adaptive thermal conductivity metamaterials: Enabling active and passive thermal control." *Journal of Thermal Science and Engineering Applications* 10.5 (2018).
- [5] Phoenix, Austin A., and Evan Wilson. "Adaptive thermal conductivity metamaterials: Enabling active and passive thermal control." *Journal of Thermal Science and Engineering Applications* 10.5 (2018).
- [6] Kasprzak, Maciej, et al. "High-temperature silicon thermal diode and switch." *Nano Energy* 78 (2020): 105261.
- [7] Hamed, Ahmed, Mahmoud Elzouka, and Sidy Ndao. "Thermal calculator." *International Journal of Heat and Mass Transfer* 134 (2019): 359-365.
- [8] Dahal, Keshab, et al. "V-VO₂ core-shell structure for potential thermal switching." *RSC advances* 7.54 (2017): 33775-33781.
- [9] Johnson, R. Wayne, et al. "The changing automotive environment: high-temperature electronics." *IEEE transactions on electronics packaging manufacturing* 27.3 (2004): 164-176.
- [10] Dreißigacker, Volker, and Sergej Belik. "High temperature solid media thermal energy storage system with high effective storage densities for flexible heat supply in electric vehicles." *Applied Thermal Engineering* 149 (2019): 173-179.
- [11] Gil, Antoni, et al. "State of the art on high temperature thermal energy storage for power generation. Part 1—Concepts, materials and modellization." *Renewable and sustainable energy reviews* 14.1 (2010): 31-55.
- [12] Liu, Ming, et al. "Review on concentrating solar power plants and new developments in high temperature thermal energy storage technologies." *Renewable and Sustainable Energy Reviews* 53 (2016): 1411-1432.

- [13] Yu, De-Hai, and Zhi-Zhu He. "Shape-remodeled macrocapsule of phase change materials for thermal energy storage and thermal management." *Applied energy* 247 (2019): 503-516.
- [14] Becattini, Viola, et al. "Experimental investigation of the thermal and mechanical stability of rocks for high-temperature thermal-energy storage." *Applied Energy* 203 (2017): 373-389.
- [15] Wang, Yilin, et al. "High temperature thermal management with boron nitride nanosheets." *Nanoscale* 10.1 (2018): 167-173.
- [16] Zhu, Huanzheng, et al. "High-temperature infrared camouflage with efficient thermal management." *Light: Science & Applications* 9.1 (2020): 1-8.
- [17] Reddy, E. Harikishan, and S. Jayanti. "Thermal management strategies for a 1 kWe stack of a high temperature proton exchange membrane fuel cell." *Applied Thermal Engineering* 48 (2012): 465-475.
- [18] Elzouka, Mahmoud, and Sidy Ndao. "High temperature near-field nanothermomechanical rectification." *Scientific reports* 7.1 (2017): 1-8.
- [19] Hamed, Ahmed, Mahmoud Elzouka, and Sidy Ndao. "Thermal calculator." *International Journal of Heat and Mass Transfer* 134 (2019): 359-365.
- [20] Li, Baowen, Lei Wang, and Giulio Casati. "Thermal diode: Rectification of heat flux." *Physical review letters* 93.18 (2004): 184301.
- [21] Yang, Nuo, et al. "Thermal rectification and negative differential thermal resistance in lattices with mass gradient." *Physical Review B* 76.2 (2007): 020301.
- [22] Dames, C. "Solid-state thermal rectification with existing bulk materials." *Journal of Heat Transfer* 131.6 (2009).
- [23] Cartoixa, Xavier, Luciano Colombo, and Riccardo Rurali. "Thermal rectification by design in telescopic Si nanowires." *Nano letters* 15.12 (2015): 8255-8259.
- [24] Wang, Yan, et al. "Phonon lateral confinement enables thermal rectification in asymmetric single-material nanostructures." *Nano letters* 14.2 (2014): 592-596.
- [25] Zhang, Teng, and Tengfei Luo. "Giant thermal rectification from polyethylene nanofiber thermal diodes." *Small* 11.36 (2015): 4657-4665.
- [26] Dettori, Riccardo, et al. "Thermal rectification in silicon by a graded distribution of defects." *Journal of Applied Physics* 119.21 (2016): 215102.
- [27] Peng, Biaolin, Huiqing Fan, and Qi Zhang. "A giant electrocaloric effect in nanoscale antiferroelectric and ferroelectric phases coexisting in a relaxor Pb0. 8Ba0. 2ZrO3 thin film at room temperature." *Advanced Functional Materials* 23.23 (2013): 2987-2992.

Reviewer #3 (Remarks to the Author):

I have reviewed this manuscript and find that it is not suitable for publication at the present stage. My main concern about this work is that the effect of the substrates on the thermal conductivity changes upon the electric field and thermal excitation in the authors' measurement is not provided to prove that this bidirectional thermal conductivity switching behavior is the intrinsic property from PZO. Although this switching behavior can be clearly demonstrated for PZO film, the bulk PZO sample does not exhibit dramatic thermal conductivity changes upon the transition region, as the authors showed in the supplementary information. The authors attributed it to existence of porosity and disorder or impurity in the bulk sample, but the measurement for a pure sample should be performed. On the other hand, the thermal conductivity measurement and the adjustment upon the electric field and thermal excitation should be performed on the substrates, and the results should be compared with that of the PZO film. These measurements would be strong evidence for confirming this observation.

Authors Response: We would like to thank the reviewer for taking the time and reviewing our manuscript. In regards to the concerns raised by the respectable reviewer for the substrate effect, we have provided additional experiments/analysis as well as discussion to demonstrate that the substrate does not play a role in our thermal conductivity measurements of PZO upon heating or biasing.

With regards to the effect of the substrate on thermal conductivity upon electrical biasing, **we should first mention that DyScO₃ (DSO) is a well-known and heavily studied *high-k* dielectric that is independent of electric field and is frequently used as a substrate for electrical measurements** [1-5]. Additionally, as depicted in the following schematic, there is no electrical field passing through the substrate that would impact its properties during our thermal conductivity measurements. In addition, DSO only transitions into a ferroelectric phase for temperatures below 4 K [6]. Due to the aforementioned reasons, we believe there is no effect from the substrate in our thermal conductivity measurement of PZO upon biasing.

Figure 4. Laser heating and electrical biasing schematic.

On the other hand, as pointed by the reviewer, the temperature could in fact change the thermal conductivity of substrate and create fictitious increase in our thermal conductivity measurements. We investigate this by separately measuring the thermal conductivity of DSO substrate as a

function of temperature. As depicted in the following plot, **the thermal conductivity of DSO substrate decreases at higher temperatures which is in great agreement with previous studies [7].** This indicates that, if the measured thermal conductivity of PZO was affected by the substrate, we should have observed a reduction in the thermal conductivity which is contrary to our experimental observation. Unlike DSO, the thermal conductivity of PZO increases with temperature and the fact that we see an increase in thermal conductivity is an indicative of the independence of our measurements from that of the substrate. This is further supported by the observation of increased thermal conductivity of the polycrystalline PZO film that was prepared on a Pt/Si substrate. Identical substrates were used in a previous study for PZT films and no spurious increases in thermal conductivity were observed [8].

Figure 5. (a) experimental data and the theoretical fit for thermal conductivity measurements at room temperature and 573 K. (b) Thermal conductivity of DSO substrate as a function of temperature. The inset shows the heat capacity of DSO estimated using equation provided in Ref. 7.

With regards to the disagreement in the measured thermal conductivity between the bulk and thin film samples, we should mention that we contacted several vendors to purchase high-quality PZO in bulk and we found it impossible to find a company that makes 100% high quality bulk PZO with no porosity and defects. Nonetheless, to investigate our hypothesis regarding the existence of porosity and impurities in our bulk PZO, we performed SEM and EDS measurements and found that **not only a significant degree of porosity exists in our measured bulk PZO as depicted in following figure but also, we found signatures of Ti impurities and non-stoichiometry.** Pure PbZrO_3 should be 20 mole percent Pb, but this ceramic was closer to 19 mole%. On the other hand, the PZO films made for this study are epitaxially grown and the balance of Pb vaporization and Pb flux during growth results in improved stoichiometry films. Furthermore, the epitaxial films have a very high degree of crystallinity and crystal perfection. In the light of these evidences, it is impossible to compare our thermal conductivity results for thin films with that of the bulk ceramics to which we have access. This discussion is added to the Supporting information.

Figure 6. PZO microstructure and nanostructure for bulk and epitaxial films.

Spectrum 1							
Element	Line Type	Apparent Concentration	k Ratio	Wt%	Wt% Sigma	At.%	Standard Label
O	K series	6.68	0.02248	14.64	0.14	61.4	SiO2
Ti	K series	0.17	0.00172	0.24	0.02	0.3	Ti
Zr	L series	23.73	0.23731	26.06	0.11	19.2	Zr
Pb	L series	60.03	0.57898	59.05	0.14	19.1	PbTe
Total:				100.00			

References

[1] Skoromets, V., et al. "Systematic study of terahertz response of SrTiO₃ based heterostructures: influence of strain, temperature, and electric field." *Physical Review B* 89.21 (2014): 214116.

[2] Kužel, P., et al. "Field-induced soft mode hardening in SrTiO₃/DyScO₃ multilayers." *Applied physics letters* 93.5 (2008): 052910.

[3] Gao, Ran, et al. "Ferroelectricity in Pb_{1+δ}ZrO₃ thin films." *Chemistry of Materials* 29.15 (2017): 6544-6551.

[4] Biegalski, Michael D., et al. "Influence of anisotropic strain on the dielectric and ferroelectric properties of SrTiO₃ thin films on DyScO₃ substrates." *Physical Review B* 79.22 (2009): 224117.

[5] Chan, K. C., et al. "Memory characteristics and the tunneling mechanism of Au nanocrystals embedded in a DyScO₃ high-k gate dielectric layer." *Semiconductor Science and Technology* 26.2 (2010): 025015.

[6] Haeni, J. H., et al. "Room-temperature ferroelectricity in strained SrTiO₃." *Nature* 430.7001 (2004): 758-761.

[7] Hidde, Julia, et al. "Thermal conductivity of rare-earth scandates in comparison to other oxidic substrate crystals." *Journal of Alloys and Compounds* 738 (2018): 415-421.

[8] Foley, Brian, and Hopkins, Patrick (advisor). *Dynamic Control of Thermal Transport Via Phonon Scattering At Ferroelastic Domain Walls*. University of Virginia, Mechanical and Aerospace Engineering - School of Engineering and Applied Science, PHD (Doctor of Philosophy), 2016, doi.org/10.18130/V3PS4Z.

My other comments I made while reading through this manuscript are listed below:

1. The temperature unit should be unified to either degree C or Kelvin.

Authors Response: We changed all the temperature units in our manuscript to Kelvin for consistency.

2. What is the measurement uncertainty for the film thermal conductivity? How was the error bars calculated in figure 3.

Authors Response: The uncertainty in Fig. 3 a,b (now Fig. 4 a,b) is calculated based on standard deviation across multiple measurements at different spots. The measurements in Fig. 3c however were performed at a single spot and its uncertainty is calculated based on 10% variations in the transducer thickness. According to our sensitivity calculations for a 2-layer model (considering PZO as an interfacial resistance), after the transducer specific heat, its thickness has the highest sensitivity in our measurements as presented in the following plot:

Figure 7. Sensitivity analysis for a 2-layer model SRO/DSO and treating the PZO as an interfacial resistance.

Since the heat capacity is a material property and would not significantly change, we assumed 10% variation in the transducer thickness. We added the following statements to the Fig. 3 caption:

“The uncertainty in a, b is based on standard deviation across multiple scans.”

“The uncertainty in c is calculated based on 10% variations in the transducer thickness.”

3. In figure 3(b), the thermal conductivity data in the transition should be presented in a inset to clearly see the thermal conductivity jump in this region.

Authors Response: We thank the reviewer for the change suggested for Fig. 3b. However, since only one of our thermal conductivity measurements as a function of temperature shows a jump in thermal conductivity near the Curie temperature, we repeated the measurement on the same sample and observed that, contrary to our initial measurements, the thermal conductivity gradually increases and plateaus above the Curie temperature. In order to investigate this further and pinpoint which measurement shows the accurate trend, we turned to additional characterizations using X-ray diffraction reciprocal space maps. collected over a temperature range of 25 to 250 °C for epitaxial PbZrO_3 on DyScO_3 . According to these measurements, we observe that the orthorhombic to cubic phase transformation is not a sudden transition but a gradual change. As can be seen in Fig. 7, the PbZrO_3 reflection clearly shows that the orthorhombic phase gradually disappears as the temperature increases to 250 °C rather than a sudden transition from 200 to 250 °C. These data not only confirm that the antiferroelectric to paraelectric transformation in epitaxial PbZrO_3 occurs over the phase transition temperature, but also indicate that this transition is gradual.

Figure 8. X-ray diffraction reciprocal space maps collected over a temperature range of 25 to 250 °C for epitaxial PbZrO_3 on DyScO_3 . (a)-(e) show maps collected about the DyScO_3 332 reflection and (f)-(j) show maps collected for the PbZrO_3 450 $^\circ$ reflection. The 450 $^\circ$ reflection disappears between 200 and 250 °C while the 440 $^\circ$ and 280 $^\circ$ reflections merge to a single peak above 150 °C. These data confirm that the antiferroelectric to paraelectric transformation in epitaxial PbZrO_3 occurs between 200 and 250 °C. The streak denoted as the PEEK dome in (f)-(j) is a reflection from the protective dome on the hot stage and not part of the sample stack.

Figure 9. Thermal conductivity of PZO deposited via different techniques with various transducer as a function of temperature. **The solid blue squares are the repeated measurements.**

4. In figure 4(b), the thermal conductivity remains almost no change upon 20 mW heating power, indicating the sample's temperature is below the transition region, while for the increase of thermal conductivity with heating power of 40 and 60 mW, the sample's temperature should be over the transition temperature. What is the approximate temperature of the sample in the heating power of 20, 40 and 60 mW?

Authors Response: In order to calculate the steady-state temperature rise due to laser heating, we use a script developed in our previous work [1]. The temperature rise in this case largely depends on the absorption coefficient of the SrRuO_3 at wavelength of 532 nm, the beam size (12 μm), and the delivered power. According to the literature, for 100 nm SrRuO_3 the absorption is near 80% [2]. Although the absorption of SrRuO_3 depends on the deposition process and the quality of the film, we take 80% as the upper limit for the absorption of SrRuO_3 . Furthermore, the measured power is before the laser passes a few mirrors and the objective. Assuming 10% power drop from the measured point to the surface of the sample, the estimated temperature rises within the probed

region for the 20, 40, and 60 mW is approximately 135, 271, and 406 K (see Fig. 9). This agrees with the observed trend in Fig. 4(b) in the main manuscript. Since the beam profile is Gaussian, there is a Gaussian temperature rise on the surface of the sample. This results in a formation of a temperature gradient in the in-plane direction. This discussion is added to the Supporting information.

Figure 10. Temperature rise profile as a function of **probe beam radius** due to a Gaussian CW heater beam at different powers.

Reference

- [1] Braun, Jeffrey L., et al. "On the steady-state temperature rise during laser heating of multilayer thin films in optical pump–probe techniques." *Journal of Heat Transfer* 140.5 (2018).
- [2] Lee, Sungki, Brent A. Apgar, and Lane W. Martin. "Strong Visible-Light Absorption and Hot-Carrier Injection in TiO₂/SrRuO₃ Heterostructures." *Advanced Energy Materials* 3.8 (2013): 1084-1090.

5. The switching thermal conductivity behavior observed in the PZO film reported in the work should also be compared with that of other related ferroelectric materials reported in literature.

Authors Response: we thank the reviewer for this suggestion. However, we have provided a thorough discussion regarding the reported thermal conductivity for the ferroelectric materials in the literature in our introduction:

“The thermal conductivity of these FE solids is dominated by phonons and changes in the domain wall density could potentially impact their scattering rate and lead to changes in the thermal conductivity. In this regard, it has been shown than in ferroelectric BiFeO₃ the thermal boundary conductance between the domains is lower than the that of the grain boundaries and result in strong scattering of vibrational modes [14]. Later, Ihlefeld et al. [5] showed the thermal conductivity of PbZr_{0.3}Ti_{0.7}O₃ / PbZr_{0.7}Ti_{0.3}O₃ bilayers, deposited on silicon substrates moderately decreases by 11% upon application of electric field; this observation could be attributed to an increase in

domain wall density and a corresponding increase in the phonon-scattering rate. In a different approach, Foley et al. [15] showed the thermal conductivity of suspended PZT membranes could be increased by 13% with electric field biasing. In this geometry, the ferroelectric film is not mechanically clamped to the substrate, which allows the domain size to increase and reduce the phonon-boundary scattering rate. In another work, Langenberg et al. [10] showed thermal conductivity of epitaxially grown PTO with high domain wall density is 61% lower than that of the single-domain film. More recently, through first-principles simulations, Liu et al. [16] showed that the thermal conductivity of PTO can be bidirectionally tuned by applying electric fields of opposite polarities.”

REVIEWERS' COMMENTS

Reviewer #1 (Remarks to the Author):

The authors have already added more details to remove my concern. This manuscript is acceptable at present in my opinion.

Reviewer #2 (Remarks to the Author):

I appreciate the respectful response to my remark. I realized that the thermal device operating at the high-temperature is justified on the certain applications. However, the description in the response, "the first experimental demonstration of a bidirectional solid-state thermal conductivity switch" is obviously wrong, because general crystalline and glassy solid show the temperature dependence in the thermal conductivity; cooling and heating could change the thermal conductivity bi-directionally. Therefore, the new finding of the temperature dependence of the thermal conductivity of PbZrO₃ film during the phase transition should be described in detail, and the highlights in the page 6 is highly appreciated. I recommend publishing this revision as is.

Reviewer #3 (Remarks to the Author):

The authors have already revised their manuscript according to my comments, and therefore I recommend it to be published in this journal.